# Biofilm-associated metabolism via *ERG251* in *Candida albicans*

**Liping Xiong[1], Nivea Pereira De Sa[2], Robert Zarnowski[3], Manning Y. Huang[4¤], Caroline Mota Fernandes[2], Frederick Lanni[4], David R. Andes[3], Maurizio Del Poeta[2], Aaron P. Mitchell**[1,4]*

**1** Department of Microbiology, University of Georgia, Athens, Georgia, United States of America, **2** Department of Microbiology and Immunology, Stony Brook University, Stony Brook, New York, United States of America, **3** Department of Medical Microbiology and Immunology, University of Wisconsin, Madison, Wisconsin, United States of America, **4** Department of Biological Sciences, Carnegie Mellon University, Pittsburgh, Pennsylvania, United States of America

¤ Current Address: Department of Biochemistry and Biophysics, University of California San Francisco, San Francisco, California, United States of America

* Aaron.Mitchell@uga.edu

**Data Availability Statement:** The authors confirm that all data underlying the findings are fully available without restriction. All relevant data are within the paper and its files.

## Abstract

Biofilm formation by the fungal pathogen *Candida albicans* is the basis for its ability to infect medical devices. The metabolic gene *ERG251* has been identified as a target of biofilm transcriptional regulator Efg1, and here we report that *ERG251* is required for biofilm formation but not conventional free-living planktonic growth. An *erg251Δ/Δ* mutation impairs biofilm formation *in vitro* and in an *in vivo* catheter infection model. In both *in vitro* and *in vivo* biofilm contexts, cell number is reduced and hyphal length is limited. To determine whether the mutant defect is in growth or some other aspect of biofilm development, we examined planktonic cell features in a biofilm-like environment, which was approximated with sealed unshaken cultures. Under those conditions, the *erg251Δ/Δ* mutation causes defects in growth and hyphal extension. Overexpression in the *erg251Δ/Δ* mutant of the paralog *ERG25*, which is normally expressed more weakly than *ERG251*, partially improves biofilm formation and biofilm hyphal content, as well as growth and hyphal extension in a biofilm-like environment. GC-MS analysis shows that the *erg251Δ/Δ* mutation causes a defect in ergosterol accumulation when cells are cultivated under biofilm-like conditions, but not under conventional planktonic conditions. Overexpression of *ERG25* in the *erg251Δ/Δ* mutant causes some increase in ergosterol levels. Finally, the hypersensitivity of *efg1Δ/Δ* mutants to the ergosterol inhibitor fluconazole is reversed by *ERG251* overexpression, arguing that reduced *ERG251* expression contributes to this *efg1Δ/Δ* phenotype. Our results indicate that *ERG251* is required for biofilm formation because its high expression levels are necessary for ergosterol synthesis in a biofilm-like environment.

## Author summary

Microbial growth in a surface-bound community or biofilm enables infection in varied contexts. For the fungal pathogen *Candida albicans*, much is known about the structural and molecular determinants of biofilm formation, as well as their biofilm-associated

**Funding:** This work was supported by NIH/NIAID grants (R01 AI146103 to APM, R01 AI073289 to DRA, R01 AI125770 to MDP), and by startup funds from the University of Georgia (to APM). The funders had no role in study design, data collection and analysis, decision to publish, or preparation of the manuscript.

**Competing interests:** The authors have declared that no competing interests exist.

regulation. Less is known about metabolic features that may support biofilm growth but are not necessary for conventional growth as free-living cells. Here we focused on a metabolic gene, *ERG251*, because it is under control of a biofilm master regulator in multiple *C. albicans* isolates. We find that *ERG251* is required for growth in biofilm-like conditions, but not in conventional free-living conditions. Biofilm-associated metabolic genes like *ERG251* may be useful therapeutic targets for eradication of biofilm infections.

## Introduction

*Candida albicans* is a prominent fungal pathogen [1,2] within the WHO critical priority group. One of its key virulence attributes is the ability to form biofilms on implanted medical devices, which leads to disseminated *Candida* infection [3–5]. Our goal is to understand the genetic determinants of *C. albicans* biofilm formation to define processes for potential therapeutic intervention.

Many features of *C. albicans* biofilm development are well understood. Hyphal cells, which are required for biofilm formation under most circumstances, express cell surface proteins such as Als1, Als3, Hwp1, and Hyr1. These proteins contribute to biofilm integrity, and mediate cell-cell or cell-substrate adherence, or both [6–10]. Biofilm cells produce extracellular matrix, comprising a mannan-glucan complex, which provides biofilm-associated drug resistance through a sponging mechanism [11]. Matrix biogenesis utilizes components delivered by extracellular vesicles [12]. *C. albicans* biofilms release yeast cells throughout development, and the dispersed cells have novel properties that distinguish them from typical yeast cells grown in planktonic culture [13]. Overall biofilm development is orchestrated by a group of transcription factors or "master regulators" that control biofilm-associated gene expression [14]. These aspects of *C. albicans* biofilm development have been reviewed extensively (see [15–20]).

One aspect of *C. albicans* biofilm development that is less well understood at this time is biofilm-associated metabolism. We use this term to describe metabolic reactions or pathways that are required for biofilm growth, but not for planktonic growth under conventional conditions. One driver of biofilm metabolism is thought to be hypoxia. For example, gene expression responses to hypoxia and to biofilm growth overlap considerably [21,22]. In addition, the biofilm master regulator Efg1 activates a subset of hypoxia-related genes [23–25]. The glycolytic regulator Tye7 offers a functional correlation: it acts to suppress hypha formation during both hypoxic growth and biofilm growth [26]. Finally, the hyper-biofilm forming *ssn3Δ/Δ* mutant has altered levels of over 200 diverse metabolites compared to the wild type [27], some of which may drive the hyper-biofilm phenotype. These reports point to *C. albicans* biofilm-associated metabolism as a promising area for study.

We have used gene expression-based approaches to find genes that function in *C. albicans* biofilm formation. One focus has been Efg1, the first transcription factor shown to be required for biofilm formation [28] and one of the most well-studied transcription factors of *C. albicans* [29]. Efg1 is required for biofilm formation in multiple *C. albicans* clinical isolates [30,31], a feature that distinguishes it from other biofilm master regulators [30]. Efg1 controls expression of ~800 genes in any one strain [30,31], a lot to study effectively. However, there is limited overlap among Efg1-regulated genes among clinical isolates [30,31], due in part to differences in expression of partner proteins that shape Efg1 outputs [32]. Only 110 genes are under Efg1 control in all 17 isolates we examined [31], a more manageable number of genes for functional analysis.

This study focuses on *ERG251*, an ergosterol biosynthetic gene. Erg251 has been of interest recently as the target of inhibitor CZ66 [33], which alters the cell's response to the ergosterol inhibitor fluconazole [33]. It has also been connected to drug tolerance, drug resistance, and virulence in a manuscript by Zhou et al. [34]. We were intrigued by *ERG251* because it is under Efg1 control [30,31]: among 17 clinical isolates, RNA levels for *ERG251* are significantly downregulated in each *efg1*Δ/Δ mutant compared to its respective wild type. This observation suggested to us that *ERG251* may have a role in biofilm formation. We explore that hypothesis here.

## Results

### Positive role of *ERG251* in biofilm formation

To explore the hypothesis that *ERG251* may function in biofilm formation, we constructed an *erg251*Δ/Δ mutant and complemented derivative in the wild-type strain SC5314. Biofilm formation was tested at 37˚C in five media (Fig 1A, 1B and 1C). In each medium, the wild type produced a robust biofilm and the mutant produced a biofilm of reduced depth and density

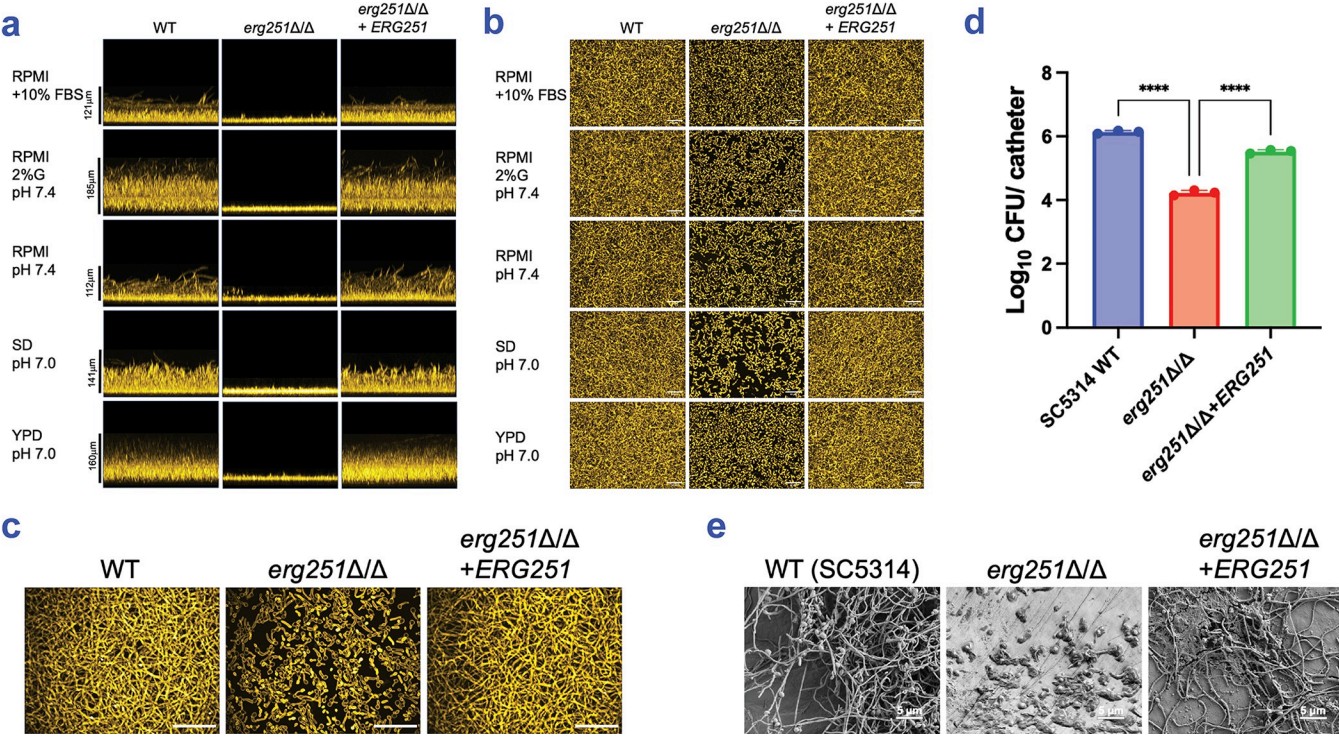

**Fig 1. Role of *ERG251* in biofilm formation.** (a) Biofilm side-view projections. *C. albicans* SC5314 wild type, *erg251*Δ/Δ mutant, and complemented strain *erg251*Δ/Δ+*ERG251* were assayed for biofilm formation under *in vitro* growth conditions. Strains were grown in RPMI+10% FBS, RPMI 2% glucose (pH 7.4), RPMI (pH 7.4), SD (pH 7.0), and YPD (pH 7.0), as indicated, in a 96-well plate at 37˚C for 24 hours. Fixed biofilms were stained with calcofluor white and imaged using a Keyence BZ-X800E fluorescence microscope. Representative projections from each biofilm are shown. Scale bars are as indicated. (b) Biofilm apical-view projections. Apical views of representative sections were generated with the same datasets used in (a). White scale bars indicate 50 μm in length. (c) Confocal microscopy. Strains were grown in RPMI+10% FBS in a 96-well plate at 37˚C for 24 hours. Fixed biofilms were stained using Concanavalin A, Alexa Fluor 594 conjugate then imaged by confocal microscopy. White scale bars indicate 50 μm in length. (d) *In vivo* biofilm formation. SC5314 wild type, *erg251*Δ/Δ mutant, and complemented strain *erg251*Δ/Δ+*ERG251* were tested for *in vivo* biofilm formation in a rat venous catheter infection model [56]. *C. albicans* cell counts per catheter were determined at 48 hours post-infection. Calculated $Log_{10}$CFU per catheter was plotted using GraphPad Prism 10 software. Mean values from three independent experiments are shown, and error bars represent the standard error of the mean (SEM). Statistical analysis was performed using one-way ANOVA, and asterisks indicate statistically significant differences. **** p-value < 0.0001. (e) Biofilm formation within the catheter lumen was assessed by scanning electron microscopy. Representative sections from each biofilm are shown. White scale bars indicate 5 μm in length. Numerical data may be found in S1 Table.

(Fig 1A and 1B). The mutant biofilms also appeared to have reduced hyphal content compared to wild type biofilms (Fig 1B), a conclusion verified for RPMI+FBS medium with high-resolution imaging (Fig 1C). The complemented strain produced biofilms comparable to the wild type in depth, density, and hyphal content (Fig 1A, 1B and 1C). These results indicate that *ERG251* is required for biofilm formation *in vitro*.

To test whether the *erg251Δ/Δ* biofilm defect is strain-specific, we constructed *erg251Δ/Δ* mutations in two additional backgrounds: P76067 and P57055. Biofilms were cultivated under our standard conditions for multi-strain analysis (RPMI+10% FBS, 37˚C, 24 hours). Each mutant presented a severe defect in biofilm depth and density compared to its respective wild type (S1 Fig). Therefore, *ERG251* is required for biofilm formation *in vitro* in several *C. albicans* strains.

To assess prospective *ERG251* clinical relevance, we determined whether *ERG251* is required for biofilm formation *in vivo* with a rat venous catheter biofilm model [35]. Comparison of infected catheters showed that the SC5314 *erg251Δ/Δ* mutant produced significantly fewer biofilm cells than either the corresponding wild type or complemented strain (Fig 1D). Catheter imaging showed that *erg251Δ/Δ* mutant cells produced germ tubes and short hyphae, while the wild type and complemented strain produced long hyphae (Fig 1E). Therefore, *ERG251* is required for biofilm formation *in vivo* and *in vitro*; it acts in each case to promote overall biofilm population size and hyphal growth.

## Environmentally contingent role of *ERG251* in growth and filamentation

A simple explanation for the *erg251Δ/Δ* phenotype is that the mutation causes a partial growth defect, which in turn affects both cell number or density and hyphal length in biofilms. We conducted growth assays under planktonic conditions (i.e., vigorously shaken cultures) to test this explanation. The *erg251Δ/Δ* mutant presented a mild defect in growth rate (Fig 2A) and yield (Fig 2B) compared to the wild type and complemented strain. We also conducted filamentation assays under planktonic conditions. The *erg251Δ/Δ* mutant produced abundant hyphae (Fig 2E) that were comparable in length (Fig 2F) to the wild type and complemented strain. The lack of a pronounced mutant defect in planktonic growth or filamentation seems at odds with the prominent mutant biofilm defect described above.

Biofilm growth conditions differ from planktonic growth conditions in several ways, including limited mixing of media surrounding the cell community. Limited mixing may lead to local accumulation of secreted metabolites and restricted exchange of oxygen, $CO_2$, and other gases. We considered the possibility that mutant defects may be most evident under biofilm-like growth conditions. Therefore, we repeated growth and filamentation assays under biofilm-like growth conditions, which we approximated with sealed, static culture vessels. Under biofilm-like conditions, the *erg251Δ/Δ* mutant presented a severe defect in growth rate (Fig 2C) and yield (Fig 2D) compared to the wild type and complemented strain. In addition, the *erg251Δ/Δ* mutant produced few hyphae (Fig 2E), with overall hyphal length severely reduced (Fig 2F), compared to the wild type and complemented strain. These results indicate that the *erg251Δ/Δ* mutant phenotype depends strongly upon cultivation in biofilm-like conditions. The defects in growth and hypha production we observed under biofilm-like conditions parallel the defects in cell density and hypha production observed during growth in a true surface-associated biofilm.

## Relationship between *ERG251* and *ERG25* in biofilm-related phenotypes

*C. albicans* has two paralogs that encode a predicted C4-sterol methyl oxidase: *ERG251* and *ERG25*. We were unable to construct an *erg251Δ/Δ erg25Δ/Δ* double mutant (data not shown),

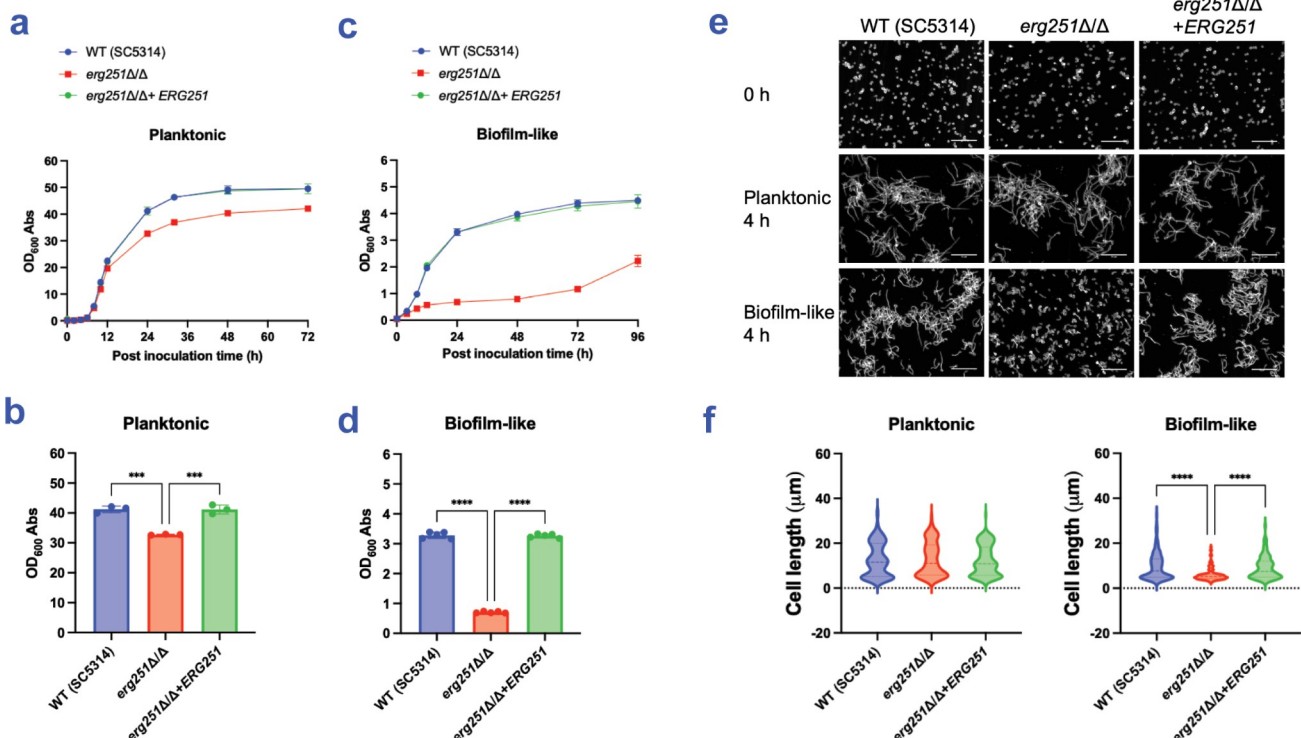

**Fig 2. Growth and filamentation assays.** (a-d) Growth assays. The SC5314 wild type, *erg251Δ/Δ* mutant, and complemented strain *erg251Δ/Δ+ERG251* were assayed for growth under planktonic (a, b) and biofilm-like (c, d) conditions. (a, c) Mean values of $OD_{600}$ of triplicates at indicated time (hours post inoculation) were plotted using GraphPad Prism 10 software; error bars represent the SEM. (b, d) Comparison of 24 hours growth yields under planktonic and biofilm-like conditions. Each measurement represents an independent biological sample. Mean values were shown, and error bars represent the SEM. Statistical analysis was performed using one-way ANOVA, and asterisks indicate statistically significant differences. *** p-value < 0.001, **** p-value < 0.0001. (e, f) Filamentation assays. Cells were assayed in RPMI+10% FBS at 37°C for 4 hours under planktonic and biofilm-like conditions, as indicated. Fixed cells were stained by calcofluor white and imaged by Zeiss fluorescence microscopy. (e) Representative images of each strain are shown. The white scale bar represents 50 μm in length. (f) Cell body lengths were quantified with a minimum of 100 cells in 3 fields of view. Hyphal units were measured between septa or between yeast cell and hyphal tip. Statistical analysis was performed using one-way ANOVA, and asterisks indicate statistically significant differences. **** p-value < 0.0001. Numerical data may be found in S1 Table.

which suggests that the two paralogs have a shared activity. The presence of two paralogs is restricted among yeasts to the CTG clade [36], which includes most pathogenic *Candida* species. Lu *et al.* showed that Erg251 protein levels are higher than Erg25 protein levels [33], and we confirmed that *ERG251* RNA levels are higher than *ERG25* RNA levels under both planktonic and biofilm-like growth conditions (S2 Fig). To determine whether *ERG25* has a biofilm-related function, an *erg25Δ/Δ* mutant was assayed for biofilm formation, growth, and filamentation in type strain SC5314. The *erg25Δ/Δ* mutant formed biofilms comparable to the wild type in five media (S3A, S3B and S3C Fig), grew similarly to the wild type under both planktonic (S4A Fig) and biofilm-like conditions (S4B Fig), and produced hyphae similarly to the wild type (S4C Fig). We confirmed the observations with *erg25Δ/Δ* mutants in the P57055 (S5 Fig) and P76067 (S6 Fig) backgrounds. Therefore, *ERG25* has little if any impact on biofilm formation, growth, and filamentation under the conditions tested.

*ERG25* may contribute less than *ERG251* to biofilm formation, growth, and filamentation because *ERG25* is expressed more weakly. This explanation predicts that overexpression of *ERG25* may relieve the need for *ERG251*. Indeed, Lu et al. found that overexpression of *ERG25* reverses the fluconazole hyper-susceptibility of an *erg251Δ/Δ* mutant [33]. To extend those results to biofilm-related phenotypes, we overexpressed *ERG25* in an *erg251Δ/Δ* background

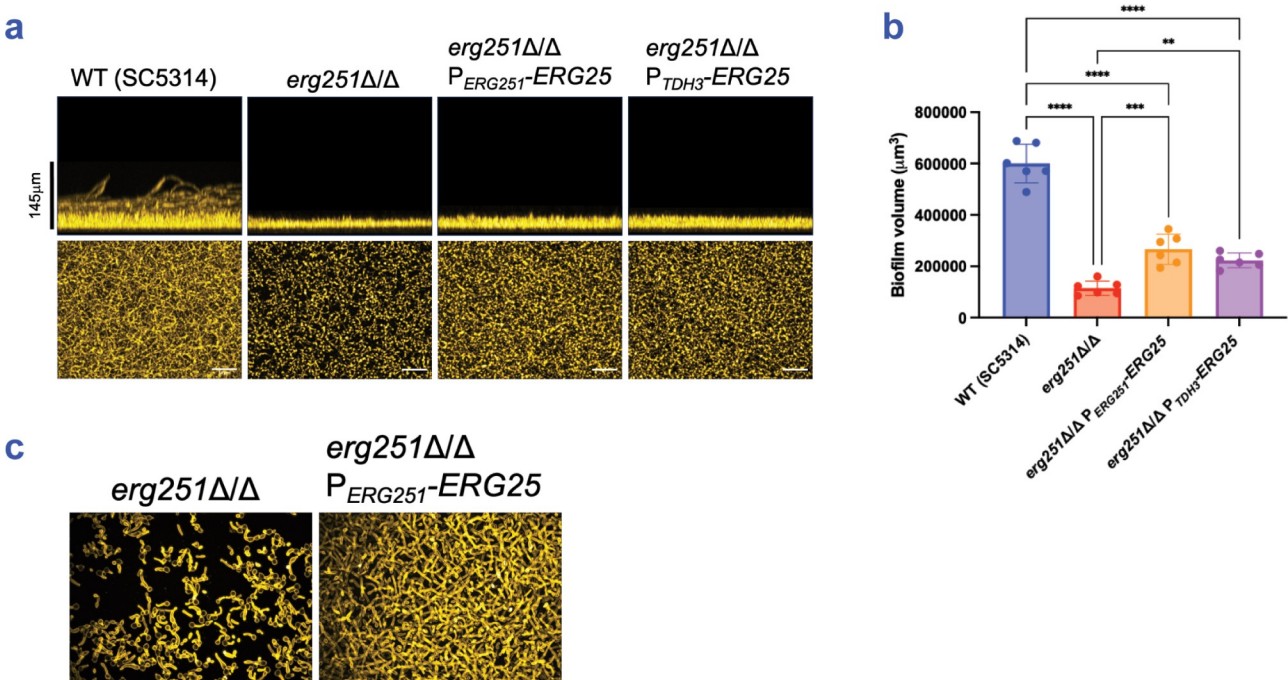

**Fig 3. Impact of *ERG25* overexpression on biofilm formation.** (a) Biofilm phenotypes. SC5314 wild type, *erg251Δ/Δ*, *erg251Δ/Δ* P$_{ERG251}$-*ERG25*, and *erg251Δ/Δ* P$_{TDH3}$-*ERG25* were assayed for biofilm formation under *in vitro* growth conditions. Strains were grown in RPMI+10% FBS in a 96-well plate at 37˚C for 24 hours. Fixed biofilms were stained with calcofluor white and imaged using a Keyence BZ-X800E fluorescence microscope. Representative side views (above) and apical views (below) are shown. Scale bars indicate depth of the wild type biofilm (above). White scale bars indicate 50 μm in length (below). (b) Biofilm volume. Volumes were measured with Image J for n = 6 biologically independent samples. Statistical analysis was performed using one-way ANOVA, and asterisks indicate statistically significant differences. ** p-value < 0.01, *** p-value < 0.001, **** p-value < 0.0001. (c) Confocal microscopy. All strains were grown in RPMI+10% FBS in a 96-well plate at 37˚C for 24 hours. Fixed biofilms were stained using Concanavalin A, Alexa Fluor 594 conjugate then imaged by confocal microscopy. White scale bars indicate 50 μm in length. Numerical data may be found in S1 Table.

by fusing the *ERG25* coding region to the promoters of *ERG251* (P$_{ERG251}$) or *TDH3* (P$_{TDH3}$). We verified *ERG25* overexpression from the P$_{ERG251}$ promoter fusion (S2C and S2D Fig). In an SC5314 *erg251Δ/Δ* background, increased *ERG25* expression resulted in a slight improvement in biofilm formation, based on biofilm depth (Fig 3A), biofilm volume (Fig 3B), and biofilm hypha production (Fig 3C). Increased *ERG25* expression also improved biofilm depth and volume by *erg251Δ/Δ* mutants in multiple media (S3 Fig) and strain backgrounds (S5 and S6 Figs). Increased *ERG25* expression improved growth of *erg251Δ/Δ* mutants under both planktonic and biofilm-like conditions (Figs 4A, 4B, S5D and S6D), and improved filamentation of *erg251Δ/Δ* mutants under biofilm-like conditions (Figs 4C, 4D, S5E, and S6E). We conclude that increased *ERG25* expression partially relieves the need for *ERG251* in all biofilm-related phenotypes examined. These results are consistent with the model that the severity of *erg251Δ/Δ* defects reflect the low *ERG25* expression levels.

## Requirement for Erg251 in ergosterol synthesis under biofilm-like conditions

We hypothesized that Erg251 is required for ergosterol synthesis during biofilm growth, but not planktonic growth. We tested this hypothesis through gas chromatography–mass spectrometry (GC-MS) analysis of sterol accumulation (Fig 5A), using SC5314-derived strains grown under planktonic or biofilm-like conditions.

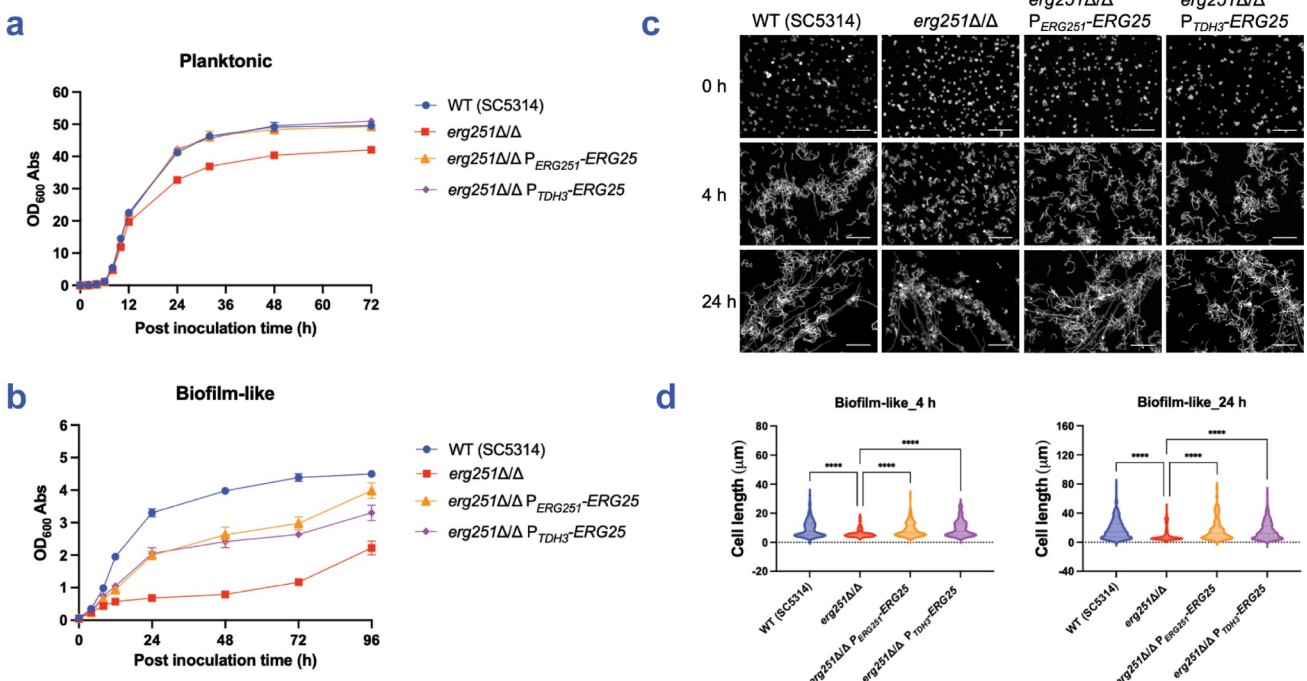

**Fig 4. Impact of *ERG25* overexpression on growth and filamentation.** (a, b) Growth phenotypes. SC5314 wild type, *erg251Δ/Δ*, *erg251Δ/Δ* P*ERG251*-*ERG25*, and *erg251Δ/Δ* P*TDH3*-*ERG25* were assayed for growth under planktonic (a) and biofilm-like (b) conditions. Strains were grown in YPD at 30˚C. Mean values are shown for three independent biological samples, and error bars represent the SEM. (c) Filamentation phenotypes. Strains were grown in RPMI+10% FBS at 37˚C for 4 hours and 24 hours under biofilm-like conditions, respectively. Fixed cells were stained with calcofluor white and imaged by Zeiss fluorescence microscopy. Representative images of each strain are shown. White scale bars represent 50 μm in length. (d) Cell body lengths were quantified with a minimum of 100 cells in 3 fields of view. Statistical analysis was performed using one-way ANOVA, and asterisks indicate statistically significant differences. **** p-value < 0.0001. Numerical data may be found in S1 Table.

Under planktonic growth conditions, the wild type, *erg251Δ/Δ* mutant, and complemented strain all had similar levels of ergosterol (Fig 5C). We included an additional control (*erg251Δ/Δ mdr1Δ/Δ*) for the complemented strain, which had *ERG251* integrated at the *MDR1* locus; its ergosterol levels were also comparable to the wild type and the other strains (Fig 5C). These results indicate that *ERG251* is not required for ergosterol accumulation under planktonic growth conditions.

Under biofilm-like growth conditions, the wild type and complemented strain had comparable ergosterol levels (Fig 5D). However, under these conditions, the *erg251Δ/Δ* and *erg251Δ/Δ mdr1Δ/Δ* strains had significantly reduced levels of ergosterol compared to the wild type and complemented strain (Fig 5D). These results indicate that *ERG251* is required for ergosterol accumulation under biofilm-like growth conditions.

We tested the role of Erg25 under both planktonic and biofilm-like growth conditions as well. Under both conditions, the wild type and *erg25Δ/Δ* mutant had comparable ergosterol levels (Fig 5C and 5D). Under planktonic conditions, the *erg251Δ/Δ* and *erg251Δ/Δ* P*TDH3*-*ERG25* strains had comparable ergosterol levels (Fig 5C). However, under biofilm-like conditions, the *erg251Δ/Δ* P*TDH3*-*ERG25* strain had slightly elevated ergosterol levels compared to the *erg251Δ/Δ* mutant (Fig 5D). These results indicate that Erg25 is not required for ergosterol accumulation under either condition, and that increased expression of *ERG25* can partially relieve the need for *ERG251* for ergosterol accumulation.

Under biofilm-like conditions, the GC-MS profiles of *erg251Δ/Δ* mutant strains included two sterol peaks that we call Sterol A and Sterol B, which eluted at 24.7 min and 25.65 min,

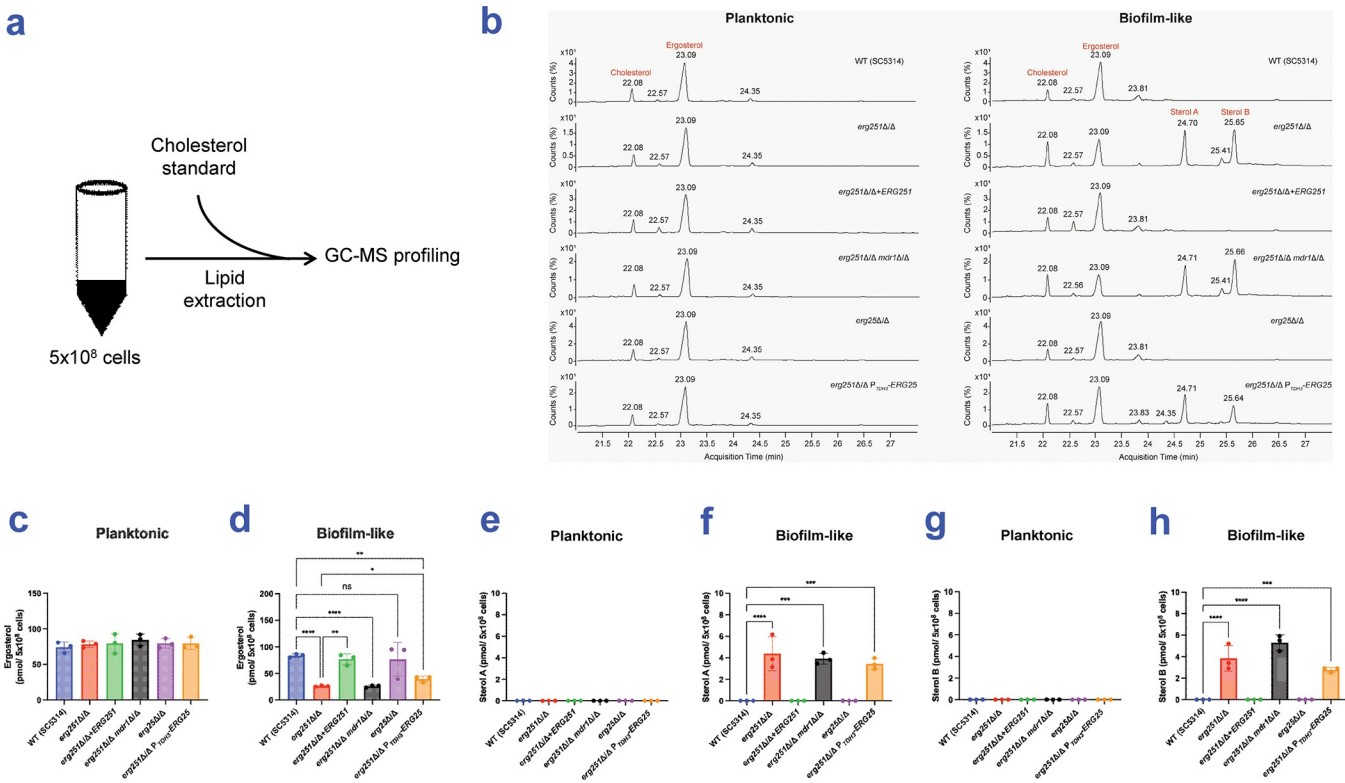

**Fig 5. GC-MS sterol analysis.** (a) Simplified workflow of lipid assay. Lipid analysis was conducted in 6 strains, including SC5314 wild type, *erg251Δ/Δ*, *erg251Δ/Δ+ERG251*, *erg251Δ/Δ mdr1Δ/Δ*, *erg25Δ/Δ*, and *erg251Δ/Δ* $P_{TDH3}$-*ERG25*. Strains were grown in CSM medium at 30˚C for 24 hours under planktonic and biofilm-like conditions, respectively. Lipids were extracted from 5x10$^8$ cells of three independent biological samples per strain and profiled by GC-MS. (b) Representative GC-MS profiling of six strains grown under planktonic and biofilm-like conditions, respectively. Numbers above peaks represent the retention time of the peak based on the number of counts taken by the mass spectrometer detector at the point of retention. Labelled peaks indicate the input standard cholesterol, ergosterol, and two unidentified sterols: Sterol A and Sterol B. (c, d) Ergosterol content per 5x10$^8$ cells in examined strains grown under planktonic (c) or biofilm-like (d) conditions. (e, f) Sterol A content per 5x10$^8$ cells in examined strains grown under planktonic (e) or biofilm-like (f) conditions. (g, h) Sterol B content per 5x10$^8$ cells in examined strains grown under planktonic (g) or biofilm-like (h) conditions. Statistical analysis was performed using T test. * p-value < 0.05, ** p-value < 0.01, *** p-value < 0.001, **** p-value < 0.0001. Numerical data may be found in S1 Table.

respectively (Fig 5B and 5E–5H). Their elution profiles did not match eburicol (28.55 min) or 4,4-dimethyl zymosterol (24.58 min), and we could not assign a structure with high confidence from the NIST database. These products were evident in the *erg251Δ/Δ* $P_{TDH3}$-*ERG25* strain as well. These sterols may be derivatives of eburicol or other intermediates that accumulate when Erg251 function is impaired.

## *EFG1*, fluconazole susceptibility, and *ERG251*

Efg1, the positive regulator of *ERG251*, is required for biofilm formation [8,28]. Efg1 is also required for normal susceptibility to the ergosterol inhibitor fluconazole [37,38]. Specifically, an *efg1Δ/Δ* mutant is hyper-susceptible to fluconazole [37,38], a finding we replicated in multiple strain backgrounds (Fig 6). We hypothesized that reduced expression of *ERG251* in *efg1Δ/Δ* mutants may cause this phenotype. To test this hypothesis, *ERG251* expression was rendered Efg1-independent by fusion to the iron-responsive *RBT5* promoter [39], and its effects were tested in low-iron RPMI medium. In *efg1Δ/Δ* mutants, $P_{RBT5}$-*ERG251* overcame fluconazole hyper-susceptibility (Fig 6). In addition, $P_{RBT5}$-*ERG251* improved growth of *EFG1+/+* wild type strains in the presence of fluconazole (Fig 6). These results indicate that reduced expression of *ERG251* contributes to the fluconazole hyper-susceptibility of *efg1Δ/Δ* mutants.

**Fig 6. Fluconazole susceptibility.** Wild type, *efg1Δ/Δ*, P*RBT5*-*ERG251*, and *efg1Δ/Δ* P*RBT5*-*ERG251* in three strain backgrounds were assayed for fluconazole susceptibility. Strains were grown in YPD at 30°C for 16 hours with shaking, then dilutions of $5^{-2}$ to $5^{-6}$ $OD_{600} \sim 3$ were spotted in RPMI (pH 7.4) agar with and without fluconazole (5 µg/ml). Plates were incubated at 37°C, images were taken between 48 and 72 hours.

## Discussion

Biofilm formation by *C. albicans* is significant as the basis for medical device-associated infection. Many aspects of *C. albicans* biofilm formation have been extensively studied; metabolic adaptations that enable biofilm growth have received less attention. Here we have shown that the

metabolic gene *ERG251* is required for biofilm formation in multiple strains and settings. Our results show that *ERG251* is required for ergosterol synthesis under biofilm-like conditions, but not under typical planktonic conditions. Overexpression of the weakly expressed paralog *ERG25* partially relieves the need for *ERG251* in support of biofilm-related phenotypes and ergosterol accumulation. *ERG251* is under positive control by the biofilm master regulator Efg1, which has been known to influence azole drug susceptibility. Overexpression studies in multiple strains argue that Efg1 regulation of *ERG251* expression contributes to this role. This study sheds light on *ERG251* as a biofilm-associated metabolic gene. Many genes are required for biofilm development, including some metabolic genes (discussed below). To our knowledge, though, *ERG251* is the only metabolic gene shown to be required for growth under biofilm conditions.

### *ERG251* function in biofilm formation

*ERG251* is required for biofilm formation *in vitro* in a variety of media and strains, and *in vivo* in a catheter infection model. The feature of the mutant that most surprised us initially was the lack of a prominent growth or filamentation defect under conventional planktonic growth conditions. In one way that feature made sense: the paralog *ERG25* may logically be able to substitute for *ERG251*. However, this phenotypic feature did not provide a simple explanation for the biofilm defect. Under assay conditions that mimic a biofilm environment, *erg251Δ/Δ* mutants of multiple strains have evident growth and filamentation defects. Those defects correlate with the mutants' biofilm defects. The phenotypic correlation between biofilm-like growth and actual biofilm formation is extended by results of *ERG25* overexpression, which partially relieves the mutant defects in biofilm formation, filamentation, and growth under biofilm-like conditions.

Ergosterol accumulation assays correlate with biofilm formation ability as well. The *erg251Δ/Δ* mutant has normal ergosterol levels after growth in conventional planktonic conditions (i.e., its levels are comparable to the wild type), but severely reduced ergosterol levels after growth in biofilm-like conditions. These assays also revealed accumulation of two unidentified sterols in the *erg251Δ/Δ* mutant, a phenotype that was unaffected by increased *ERG25* expression. These sterols may contribute to the *erg251Δ/Δ* mutant growth defect, as other sterols are known to have toxic effects [40].

Why would *ERG251* have a prominent role in biofilm growth yet a modest role in planktonic growth? One driver of this conditional phenotype is likely hypoxia. Ergosterol synthesis demands oxygen [41,42]. Under hypoxic conditions in a true biofilm or during biofilm-like growth, flux through the ergosterol pathway may be slowed by oxygen limitation. A second driver for this conditional phenotype may be limiting iron, because heme is the cofactor for multiple oxygen-requiring ergosterol biosynthetic enzymes [42]. In fact, the reaction catalyzed by Erg251 uses 3 oxygen molecules and 6 heme-containing ferrocytochrome b5 molecules (KEGG reaction R07509). High expression levels of *ERG251* would then be vital to ameliorate the impact of hypoxia and iron limitation and enable vigorous growth. According to this explanation, the defect in filamentation of *erg251Δ/Δ* mutants is a consequence of the growth impairment caused by reduced ergosterol levels. We note that subinhibitory fluconazole treatment, which has similar impact on ergosterol levels, has been shown to inhibit hypha formation [43]. Therefore, the biofilm requirement for *ERG251* can be explained most simply by its role in ergosterol synthesis (Fig 7).

### Biofilm-associated metabolism

The environment *C. albicans* encounters in a biofilm is different from what it encounters under typical planktonic growth conditions. The difference in oxygen levels is well established

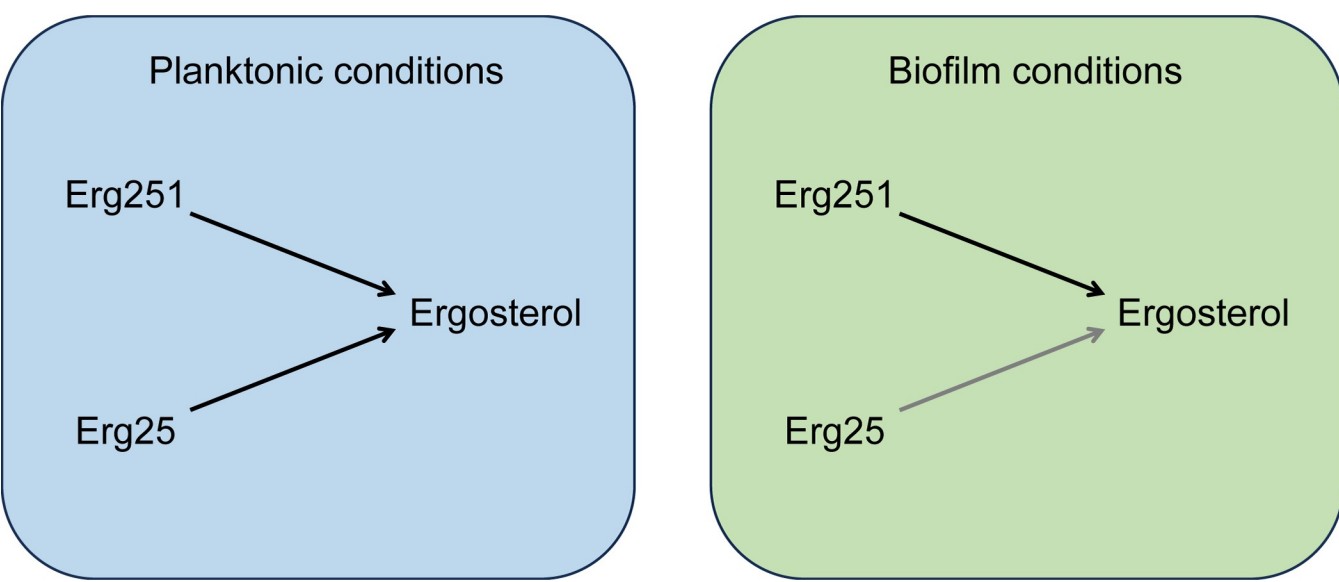

**Fig 7. Functions of *ERG251* and *ERG25* in ergosterol metabolism.** *C. albicans* has two paralogs that specify C-4 sterol methyl oxidase, *ERG251* and *ERG25*. Under planktonic conditions, either Erg251 or Erg25 is sufficient to support ergosterol synthesis. Under biofilm conditions, Erg251 is required for ergosterol synthesis at levels adequate for growth and filamentation. Biofilm conditions may impair ergosterol synthesis through limitation of oxygen and iron. *ERG251* may be more critical than *ERG25* for growth under biofilm conditions due to differences in expression levels, substrate affinities, catalytic rates, or ability to assemble into the C-4 demethylation complex [42,64] with Erg26, Erg27, and Erg28.

from overlap in both expression profiling [21,22] and mutant phenotype [26]. Other potential metabolic impacts may derive from local accumulation of metabolic byproducts such as $CO_2$; secreted organic molecules such as farnesol and tyrosol [44,45]; extracellular vesicles, which carry diverse molecular cargo [12]; and demands of extracellular matrix synthesis [17]. Metabolomic analysis is well suited to develop a global understanding of the differences between planktonic and biofilm physiology. However, one detailed study found a limited correlation between strains and conditions [46], noting that biofilm cell heterogeneity is extensive. For this reason, we consider genetic approaches to be an important complement to metabolomic strategies.

Four metabolic genes have positive roles in biofilm growth or development: *RHR2*, *CAN2*, *EHT1* [14,26,47], and *ERG251*. For *RHR2*, which specifies a putative phosphoglycerol phosphatase, and *CAN2*, which specifies a putative arginine permease, deletion mutations reduce biofilm formation [14,26,47]. For *EHT1*, which specifies a putative O-acyltransferase, a deletion mutation causes no obvious defect, but its overexpression can rescue other biofilm-defective mutants [14]. The biofilm growth medium used for *in vitro* assays undoubtedly influences metabolic mutant phenotypes. However, *RHR2* and *ERG251* are required for biofilm formation in an *in vivo* catheter infection model, a validation of their significance for pathogenesis. An additional metabolic gene has a negative role in biofilm growth: *ADH1*, which specifies an alcohol dehydrogenase, impedes biofilm growth *in vitro* and *in vivo* [48]. For each of these metabolic genes, their regulation suggested a role in biofilm growth: *RHR2* is upregulated in biofilm cells compared to planktonic cells; *ADH1* is downregulated in biofilm cells; the other three genes are under control of one or more biofilm master regulators. These simple criteria provide many metabolic genes: biofilm upregulation spotlights 18–45 lipid metabolism genes, 44–327 transport genes, and 6–32 carbohydrate metabolism genes [14,47]. There is clearly a large cohort of biofilm-regulated metabolic genes for further study.

## Methods

### Strains, media and culture conditions

A full list of strains used in this study is in S2 Table. Strain stocks were maintained in 15% glycerol at -80˚C. Prior to experiments, strains were streaked on YPD agar (2% dextrose, 2% Bacto peptone, 1% yeast extract, 2% Bacto Agar) at 30˚C for 48 hours. Overnight cultures were grown in liquid YPD medium at 30˚C with agitation. Transformants were selected on CSM-His (1.7% Difco yeast nitrogen base with ammonium sulfate with amino acid supplement lacking histidine, 2% dextrose, 2% agar) or YPD plate containing 400 μg/ml nourseothricin (clonNAT, Gold Biotechnology).

Growth assays were conducted in YPD medium at 30˚C under planktonic and biofilm-like conditions. Filamentation assays were performed in liquid RPMI-1640 medium (Sigma-Aldrich, Inc., St. Louis) adjusted to pH 7.4 with 10% fetal bovine serum (FBS) (Biotechne R&D systems S11550) at 37˚C under planktonic and biofilm-like conditions. Biofilm formation was assayed in various media, including RPMI+10% FBS, RPMI pH 7.4, RPMI with 2% glucose (pH 7.4), SD pH 7.0 (1.7% Difco yeast nitrogen base without amino acid, 2% dextrose, 25 mM HEPES), YPD pH 7.0 with and without 0.002% ergosterol. Ergosterol powder (Sigma- Aldrich PHR1512) is dissolved in 50% Tween 80 (Sigma-Aldrich P1754) 50% Ethanol (Thermo Scientific UN1170). For lipid analysis, cells were grown in liquid CSM medium (1.7% Difco yeast nitrogen base with ammonium sulfate with amino acid supplement, 2% dextrose) at 30˚C for 24 hours under planktonic or biofilm-like conditions. Fluconazole susceptibility was assayed in RPMI-1640 (pH 7.4) agar plates with fluconazole. Fluconazole powder (Sigma- Aldrich F8929) is dissolved Dimethyl sulfoxide (Sigma-Aldrich C6164).

### Strain construction

To manipulate the *C. albicans* genome, the transient CRISPR-Cas9 system was employed as previously described in detail [30,49]. All primers and plasmids [39,49–52] used in this study are in S2 Table.

To construct an *erg251Δ/Δ* mutant in wild type strain background, two halves of *ERG251* deletion cassettes were amplified from the plasmid pmh01 with primers "HIS1 CRIME/F" and "ERG251_HIS1/AR", and from the plasmid pmh02 with primers "ERG251_HIS1/AF" and "HIS1 CRIME/R", respectively. Strains MC1, MC2, and MC5 were transformed with approximately 1 μg of Cas9, 1 μg of ERG25 sgRNA, 2 μg of *HIS1_01*, and 2 μg of *HIS1_02* repair template. Transformants were screened on CSM-His media, and candidates were genotyped by PCR using primers "ERG251 CK_up/F" and "ERG251 CK_int/R" for absence of *ERG251* ORF and using primers "ERG251 CK_up/F" and "HIS1_CK_Int/R" for presence of *HIS1* marker at the *ERG25* locus. And the *NAT1* marker was recycled as described [30,49].

To construct ectopic expression strains in the SC5314 *erg251Δ/Δ* mutant, we replaced the *MDR1* ORF region with *ERG251* allele of SC5314 strain using our concatemer assembly method [53]. The *ERG251^SC5314* allele containing 2779bp of *ERG251* upstream region, the *ERG251* ORF, and 393bp of *ERG251* downstream region was amplified from SC5314 genomic DNA with primers "ERG251 Cp5'F/MDR1 up/F" and "ERG251 Cp3'R/pNAT 5'R" which are containing concatenating homology to selective marker gene, *NAT1*. The *NAT1* cassette was amplified from the plasmid pNAT with primers "pNAT_AF" and "pNAT 3'R -> MDR1 AR". Transformation was conducted with approximately 1 μg of Cas9, 1 μg of MDR1-5 sgRNA, 2 μg of *ERG251* cassette, and 2 μg of *NAT1* cassette. Transformants were screened on YPD plate containing 400 μg/ml nourseothricin, and candidate colonies were genotyped by PCR using primers "MDR1_CK_up/F" and "MDR1_CK_int/R" for absence of *MDR1* ORF and

using primers "MDR1_CK_up/F" and "ERG251 Cp Pro CK_int/R" for presence of repair template at the *MDR1* locus.

To construct an *mdr1*Δ/Δ mutant in the SC5314 *erg251*Δ/Δ mutant, the *MDR1* deletion cassette was amplified from the plasmid pNAT with primers "Mdr1_deletion_NAT_F" and "Mdr1_deletion_NAT_R" [32]. Strain *erg251*Δ/Δ was transformed with approximately 1 µg of Cas9, 1 µg of MDR1-5 sgRNA and 2 µg of *NAT1* repair template. Transformants were screened on YPD containing 400 µg/ml nourseothricin, and candidates were genotyped by PCR using primers "MDR1_CK_up/F" and "MDR1_CK_int/R" for absence of *MDR1* ORF and using primers "MDR1_CK_up/F" and "pNAT_CK_int/R" for presence of *NAT1* marker at the *MDR1* locus.

To construct an *erg25*Δ/Δ mutant in wild type strain backgrounds, two halves of *ERG25* deletion cassettes were amplified from the plasmid pmh01 with primers "HIS1 CRIME/F" and "ERG25_HIS1/AR", and from the plasmid pmh02 with primers "ERG25_HIS1/AF" and "HIS1 CRIME/R", respectively. Strains MC1, MC2, and MC5 were transformed with approximately 1 µg of Cas9, 1 µg of ERG25 sgRNA, 2 µg of *HIS1_01*, and 2 µg of *HIS1_02* repair template. Transformants were screened on CSM-His media, and candidates were genotyped by PCR using primers "ERG25 CK_up/F" and "ERG25 CK_int/R" for absence of *ERG25* ORF and using primers "ERG25 CK_up/F" and "HIS1_CK_Int/R" for presence of *HIS1* marker at the *ERG25* locus. And the *NAT1* marker was recycled as described [30,49].

To construct an *ERG25* overexpression mutant in the *erg251*Δ/Δ mutant background, the *ERG25* promoter deletion cassette was amplified from the plasmid pCJN542 with primers "ERG25_TDH3/AF" and "ERG25_TDH3/AR". An *erg251*Δ/Δ was transformed with approximately 1 µg of Cas9, 1 µg of ERG25 promoter sgRNA and 2 µg of *NAT1-TDH3* repair template. Transformants were screened on YPD containing 400 µg/ml nourseothricin, and candidates were genotyped by PCR using primers "ERG25 OE CK/F" and "ERG25 OE CK/R" for absence of *ERG25* native promoter and using primers "ERG25 Pro CK/F" and "pNAT_CK_int/R" for presence of *NAT1* marker at the *ERG25* promoter region.

To construct an *ERG25* overexpression strain using the *ERG251* native promoter, two halves of *ERG251* deletion cassettes were amplified from the from SC5314 genomic DNA with primers "ERG251 up-ERG25_OF" and "ERG25_OR/ pNAT 5'R", and from plasmid pNAT with primers "pNAT_AF" and "ERG251 dw/ pNAT AR", respectively. An *erg25*Δ/Δ mutant was transformed with approximately 1 µg of Cas9, 1 µg of *ERG251* ORF sgRNA and 2 µg of *ERG251* deletion cassettes. Transformants were screened on YPD containing 400 µg/ml nourseothricin, and candidates were genotyped by PCR using primers "ERG251 CK_up/F" and "ERG251 CK_ int/R" for absence of *ERG251* ORF, using primers "ERG251 CK_up/F" and "ERG25 CK_int/R" for presence of *ERG25* at the *ERG251* ORF region, and using primers "ERG251 CK_up/F" and "pNAT_CK_int/R" for presence of *NAT1* marker following *ERG25* fragment.

To construct ectopic expression of *ERG25* strains, we replaced the *ERG251* ORF with *ERG25* ORF allele of SC5314 wild type using our concatemer assembly method [53]. The *ERG25*<sup>SC5314</sup> allele containing 927bp *ERG25* ORF, and 488bp of *ERG25* downstream region was amplified from SC5314 genomic DNA with primers ERG251 up-ERG25_OF" and "ERG25_OR/ pNAT 5'R", and another part of *ERG251* deletion cassettes was amplified from plasmid pNAT with primers "pNAT_AF" and "ERG251 dw/ pNAT AR". Wild type strains P57055 and P76067 were transformed with approximately 1 µg of Cas9, 1 µg of *ERG251 ORF* sgRNA and 2 µg of *ERG251* deletion cassettes. Transformants were screened on YPD containing 400 µg/ml nourseothricin, and candidates were genotyped by PCR using primers "ERG251 CK_up/F" and "ERG251 CK_ int/R" for absence of *ERG251* ORF, using primers "ERG251 CK_up/F" and "ERG25 CK_int/R" for presence of *ERG25* at the *ERG25*1 ORF region, and

using primers "ERG251 CK_up/F" and "pNAT_CK_int/R" for presence of *NAT1* marker following *ERG25* fragment.

To construct an *ERG251* overexpression mutant in wild type and *efg1Δ/Δ* mutant backgrounds, the *ERG251* promoter deletion cassette was amplified from the plasmid pCJN542 with primers "ERG251_RBT5/AF" and "ERG25_RBT5/AR". Wild type strain and an *efg1Δ/Δ* were transformed with approximately 1 μg of Cas9, 1 μg of *ERG251* promoter sgRNA and 2 μg of *NAT1-RBT5* repair template. Transformants were screened on YPD containing 400 μg/ml nourseothricin, and candidates were genotyped by PCR using primers "ERG251 OE CK/F" and "ERG251 OE CK_int/R" for absence of *ERG251* native promoter and using primers "ERG251 OE CK/F" and "pNAT_CK_int/R" for presence of *NAT1* marker at the *ERG251* promoter region.

### Biofilm formation assay and imaging

Biofilm production and imaging procedures followed previous published methods with minor modifications [32,54]. To assay biofilm formation in 96-well plates, cells were grown in 5 ml of YPD overnight at 30˚C with agitation, then washed with $H_2O$ and transferred to 100 μl of pre-warmed biofilm medium (as indicated in figures) in a 96-well plate (Greiner, 655090) to achieve an $OD_{600}$ of 0.5. Cells were incubated in at 37˚C for 90 min with mild-shaking (60 rpm) to allow adherence, then each well was gently washed twice with PBS to remove non-adhered cells. One-hundred microliters of pre-warmed assay medium was added into each well, and cells were allowed to form biofilm in a shaker incubator with 60 rpm at 37˚C for 24 hours. The next day, medium was carefully discarded from each well, biofilms were fixed by incubation with 100 μl of 4% formaldehyde in PBS solution for 1 hour at room temperature (RT) and then gently washed twice with PBS. Subsequent fixed biofilms were stained with calcofluor white (200 μg/ml in PBS) overnight at room temperature with mild shaking (60 rpm), then each well was gently washed twice with PBS. For clarification and refractive index matching, biofilms were incubated with 50% of 2,2′-Thiodiethanol (TDE) in PBS for 1 hour at room temperature and then 100% TDE was added to each biofilm. Biofilms were imaged by using a Keyence fluorescence microscope with a Keyence 20X objective and 2X zoom. For confocal microscopy, fixed biofilms in a 96-well plate (Greiner/MicroClear) were stained with Concanavalin A, Alexa Fluor 594 conjugate (Life Technologies) then washed in PBS, and clarified with thiodiethanol (TDE, refractive index 1.521) as described above. Wells were imaged by using a slit-scan confocal optical unit on a Zeiss Axiovert 200M microscope with a Zeiss 25x 0.8NA multi-immersion objective, oil-immersion between objective and plate, and 1.6x internal magnification (40x total mag.) to exceed conventional Nyquist sampling. Confocal imaging was conducted as described by Lagree et al. [55]. Immediately afterward, the bottom of the plate was cleaned with 70% isopropanol on soft lens tissue to remove residual immersion oil from the MicroClear plastic. The plastic plate is unaffected by TDE in the wells.

### *In vivo Candida* vascular catheter biofilm model

*In vivo* biofilm production was performed with a rat jugular venous catheter model [56]. A $10^6$ cells/ml inoculum for each strain was allowed to grow on an internal jugular catheter placed in a pathogen-free female rat (16-week-old, 400 g) for 48 hours. After this period, the catheter volumes were removed, and the catheters were flushed with 0.9% NaCl. The catheters were then removed from the animals, and biofilms were dislodged by sonication and vortexing. Viable cell counts were determined by dilution plating. Three animal and culture replicates were used per strain.

SEM images were acquired on a Zeiss Gemini 450 scanning electron microscope with an Airlock module using an accelerating voltage of 3.0 kV, a working distance of 3 mm, an Everhart-Thornley SE2 detector with optically coupled photomultiplier, and the Zeiss SmartSEM (v. 6.05) software [57].

### Growth assay

To assay the growth phenotypes, cells were inoculated into fresh YPD medium at an initial $OD_{600}$ of 0.05. For planktonic conditions, cells were inoculated into 25 ml of YPD in 125 ml flasks and incubated at 30˚C with shaking. For biofilm-like growth, cells were inoculated into 5 ml of YPD in 30 ml glass tubes. Tubes were sealed with parafilm and extra layer of tape, then incubated at 30˚C statically.

### Filamentation assay and imaging

To assay hyphal formation in *C. albicans* strains, cell culture and fixation were performed according to previously published methods [30]. Filamentation phenotypes in RPMI (pH 7.4) with 10% FBS were assayed under both planktonic conditions (at 37˚C with shaking for 4 hours) and biofilm-like conditions (at 37˚C statically for 4 hours or 24 hours). Cells were then collected and fixed with 4% formaldehyde for 15 minutes with vortexing. After washing with PBS, fixed cells were treated by Proteinase K (Thermo Scientific Catalog number EO0491) at 37˚C (water bath) for 2 hours, then stained with calcofluor white (200 mg/ml) for 15 minutes. Stained cells were then imaged using a Zeiss Axiovert 200 fluorescence microscope. The cell length of the entire population was measured using ImageJ. At least 100 inter-septal distance measurements were taken from 800 x 800 pixels fields of view.

### RNA preparation and qPCR

RNA preparation was performed as previously described [58]. Cells were grown in 5 ml of YPD overnight at 30˚C. The next day, cells were inoculated with an initial $OD_{600}$ of 0.05 in 25 ml of YPD and grown for 24 hours under planktonic or biofilm-like conditions. Then cells were harvested by vacuum filtration and quickly frozen at -80˚C until RNA extraction. Three cultures of each strain were grown to provide three biological replicates. Harvested cells were lysed using a Bead Beater and a Qiagen RNeasy mini kit (catalog number 74104). RNA was purified with the RNeasy kit and reverse transcribed to cDNA after DNase I treatment using the iScript cDNA synthesis kit (catalog number 172–5034). Then, qPCR was performed using iQ SYBR green Supermix (catalog number 170–8880). *ERG25* and *ERG251* mRNA levels were normalized to the *ACT1* gene and compared using the threshold cycle $\Delta\Delta C_T$ method. Differences between strains were analyzed with the T-test or one-way ANOVA test.

### Lipid assay

Lipid extraction and GC-MS was conducted as previously described [59]. Pellets with $5 \times 10^8$ cells were used for lipid extraction. The dried total samples were resuspended in 100 μL chloroform added to 100 μL of BSTFA reagent (Thermo Scientific) and incubated at 70˚C for 1 hour prior to GC-MS (Agilent 7890B GC–MS, Agilent 5977A MSD) analysis [60]. The retention time and mass spectral patterns of a sterol standard were used as references for lipid analysis. Sterol standards used in this study include cholesterol (Avanti 700100), ergosterol (Cayman 19850), 4,4-Dimethyl zymosterol (Avanti 700073), Zymosterol (Avanti 700068), and Eburicol (Smolecule S633611). The relative amount of sterols A and B were estimated based

on the relative percentage of the sterol to ergosterol peak areas in each sample. Cholesterol was added as an internal standard for these analyses prior to lipid extraction.

## Fluconazole sensitivity assays

Fluconazole sensitivity was determined using a spotting assay. Cells were grown in YPD overnight at 30°C with agitation, washed with $H_2O$, diluted in $H_2O$ to an $OD_{600}$ of 3.0. Five-fold dilutions were spotted using a multichannel pipette onto RPMI agar plates with and without fluconazole (5 μg/ml). Plates were incubated at 37°C; images were taken between 48~72 hours.

## Data analysis software and statistics

Images were compiled and any adjustments were performed in ImageJ [61]. Single guide RNA sequences were checked for specificity using Cas-OFFinder software [62]. Images of biofilm and filamentation and biofilm volumes were processed using Image J (Fiji) [63]. Growth data, biofilm volume data, and qPCR data were processed by GraphPad Prism version 10 (Graph-Pad Software, Inc., La Jolla).

## Supporting information

**S1 Table. Data supplement.** This Excel file provides numerical data for Figs 1, 2, 3, 4, and 5.
(XLSX)

**S2 Table. Strains, plasmids, primers.** This Excel file provides the *C. albicans* strains and genotypes used in this study, the plasmids used in this study, and the sequences of primers used in this study.
(XLSX)

**S1 Fig. *erg251Δ/Δ* mutant biofilm phenotypes in P57055 and P76067 backgrounds.** *C. albicans* wild type P57055 and P76067, and their respective *erg251Δ/Δ* mutants were assayed for biofilm formation under *in vitro* growth conditions. Strains were grown in RPMI+10% FBS in a 96-well plate at 37°C for 24 hours. Fixed biofilms were stained with calcofluor white and imaged using a Keyence BZ-X800E fluorescence microscope. Representative sections from each biofilm are shown. Scale bars indicate depth of the corresponding biofilm by wild type strain. White scale bars of apical view images indicate 50 μm in length.
(PDF)

**S2 Fig. Gene expression of *ERG25* and *ERG251*.** (a, b) Graph indicating relative mRNA level of *ERG251* to *ERG25*. SC5314 wild type was grown in YPD at 30°C for 24 hours under planktonic (a) or biofilm-like (b) conditions. (c, d) Graph indicating relative *ERG25* mRNA levels in *ERG25* overexpression strain. *erg251Δ/Δ* mutant and *erg251Δ/Δ* $P_{ERG251}$-*ERG25* were grown in YPD at 30°C for 24 hours under planktonic (c) and biofilm-like (d) conditions. RNAs of three independent biological samples were extracted for qPCR determination. Relative gene expression was compared using the threshold cycle ΔΔCT method. Statistical analysis was performed using T test. ** p-value < 0.01, *** p-value < 0.001, **** p-value < 0.0001.
(PDF)

**S3 Fig. Biofilm phenotypes of erg25Δ/Δ mutant and ERG25 overexpression strains.** (a) Biofilm side-view projections. Five strains, including SC5314 wild type, *erg25Δ/Δ*, *erg251Δ/Δ*, $P_{ERG251}$-*ERG25 erg251Δ/Δ*, and $P_{TDH3}$-*ERG25 erg251Δ/Δ*, were assayed for biofilm formation under *in vitro* growth conditions. Strains were grown in 5 media, including RPMI+10% FBS, RPMI 2% glucose (pH 7.4), RPMI (pH 7.4), YPD (pH 7.0) + ergosterol (0.002%), and YPD (pH 7.0), in a 96-well plate at 37°C for 24 hours. Fixed biofilms were stained with calcofluor

white and imaged using a Keyence BZ-X800E fluorescence microscope. Scale bars indicate depth of the corresponding wild type biofilm. (b) Biofilm apical-view projections. Apical views of representative sections were generated with the same datasets used in (a). White scale bar indicates 50 μm in length. (c) Biofilm volume, measured with Image J and presented in column with 4 or 6 biologically independent samples. Statistical analysis was performed using one-way ANOVA. * $p$-value < 0.05, ** $p$-value < 0.01, *** $p$-value < 0.001, **** $p$-value < 0.0001. (PDF)

**S4 Fig. erg25Δ/Δ mutant growth and filamentation phenotypes.** Growth phenotypes of SC5314 wildtype and *erg25Δ/Δ* were assayed in YPD medium at 30˚C under planktonic (a) and biofilm-like (b) conditions, respectively. Mean values of $OD_{600}$ Abs of triplicates at indicated time (hours post inoculation) were plotted using GraphPad Prism 10 software. And error bars represent the SEM. (c) Filamentation phenotypes were assayed in RPMI+10% FBS at 37˚C for 4 hours under planktonic and biofilm-like conditions, respectively. Fixed ells were stained by calcofluor white and imaged by Zeiss fluorescence microscope. Representative images of each strain are shown. White scale bars indicate 50 μm in length. (PDF)

**S5 Fig. Phenotypes of P57055 wild type and its derivative strains.** (a) Biofilm side-view projections. Five strains including P57055 wild type, *erg25Δ/Δ*, *erg251Δ/Δ*, *erg251Δ/Δ* $P_{ERG251}$-*ERG25*, and *erg251Δ/Δ* $P_{TDH3}$-*ERG25* were assayed for biofilm formation under *in vitro* growth conditions. Strains were grown in three media including RPMI +10% FBS, YPD (pH 7.0) + ergosterol (0.002%), and YPD (pH 7.0) in a 96-well plate at 37˚C for 24 hours. Fixed biofilms were stained with calcofluor white and imaged using a Keyence BZ-X800E fluorescence microscope. Representative sections from each biofilm are shown. And relevant genotypes are given above each column. Scale bars indicate depth of the corresponding biofilm by wild type strain. (b) Biofilm apical-view projections. Apical views of representative sections were generated with the same datasets used in (a). White scale bars indicate 50 μm in length. (c) Biofilm volume, measured with Image J and presented in column with n = 4 biologically independent samples. Statistical analysis was performed using one-way ANOVA. ** $p$-value < 0.01, *** $p$-value < 0.001, **** $p$-value < 0.0001. (d) Growth phenotypes assayed under biofilm-like conditions. Each dot represents the average $OD_{600}$ Abs of triplicates at indicated time. (e) Filamentation phenotypes assayed under biofilm-like conditions. Strains were grown in RPMI+10% FBS at 37˚C for 4 hours. Fixed cells were stained by calcofluor white and imaged by Zeiss fluorescence microscope. Representative images of each strain are shown. White scale bars indicate 50 μm in length. (PDF)

**S6 Fig. Phenotypes of P76067 wild type and its derivative strains.** (a) Biofilm side-view projections. P76067 wild type, *erg25Δ/Δ*, *erg251Δ/Δ*, *erg251Δ/Δ* $P_{ERG251}$-*ERG25*, and *erg251Δ/Δ* $P_{TDH3}$-*ERG25* were assayed for biofilm formation under *in vitro* growth conditions. Strains were grown in three media, including RPMI +10% FBS, YPD (pH 7.0) with and without ergosterol (0.002%) in a 96-well plate at 37˚C for 24 hours. Fixed biofilms were stained with calcofluor white and imaged using a Keyence BZ-X800E fluorescence microscope. Scale bars indicate depth of the corresponding biofilm by wild type strain. (b) Biofilm apical-view projections. Apical views of representative sections were generated with the same datasets used in (a). White scale bars indicate 50 μm in length. (c) Biofilm volume, measured with Image J and presented in column with n = 4 biologically independent samples. Statistical analysis was performed using one-way ANOVA. ** $p$-value < 0.01, *** $p$-value < 0.001, **** $p$-value < 0.0001. (d) Biofilm-like growth phenotypes. Each dot represents the average $OD_{600}$

Abs of triplicates at indicated time. (e) Biofilm-like filamentation phenotypes. Strains were grown in RPMI +10% FBS at 37˚C for 8 hours. Fixed ells were stained by calcofluor white and imaged by Zeiss fluorescence microscope. Representative images are shown. White scale bar represents 50 μm in length.
(PDF)

## Acknowledgments

We are grateful to Max Kuhr for exceptional lab management and technical support, to all past and current Mitchell lab members for their continued interest and many helpful discussions, and to Drs. Xiaorong Lin, Douda Benssason, and Michelle Momany, for advice and support. We thank Dr. John Haley from Stony Brook Proteomics and Lipidomics facility who provided valuable assistance for GC-MS analysis. We are extremely grateful to the Selmecki and Cowen labs for sharing results prior to publication, and to Drs. Anna Selmecki and Laura Burrack for their comments on this manuscript.

## Author Contributions

**Conceptualization:** Liping Xiong, Manning Y. Huang, David R. Andes, Maurizio Del Poeta, Aaron P. Mitchell.

**Data curation:** Nivea Pereira De Sa.

**Formal analysis:** Liping Xiong, Nivea Pereira De Sa, Robert Zarnowski, Manning Y. Huang, Frederick Lanni.

**Funding acquisition:** Frederick Lanni, David R. Andes, Maurizio Del Poeta, Aaron P. Mitchell.

**Investigation:** Liping Xiong, Robert Zarnowski, Caroline Mota Fernandes, Frederick Lanni.

**Methodology:** Liping Xiong, Nivea Pereira De Sa, Robert Zarnowski, Manning Y. Huang.

**Project administration:** David R. Andes, Maurizio Del Poeta, Aaron P. Mitchell.

**Resources:** Liping Xiong, Manning Y. Huang.

**Supervision:** David R. Andes, Maurizio Del Poeta, Aaron P. Mitchell.

**Writing – original draft:** Liping Xiong, Aaron P. Mitchell.

**Writing – review & editing:** Nivea Pereira De Sa, Frederick Lanni, David R. Andes, Maurizio Del Poeta, Aaron P. Mitchell.

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
