## [Decision Letter · Decision Letter 0]

18 Mar 2024

Dear Professor Mitchell,

Thank you very much for submitting your manuscript "Biofilm-associated metabolism via ERG251 in Candida albicans" for consideration at PLOS Pathogens. As with all papers reviewed by the journal, your manuscript was reviewed by members of the editorial board and by several independent reviewers. The reviewers appreciated the attention to an important topic. Based on the reviews, we are likely to accept this manuscript for publication, providing that you modify the manuscript according to the review recommendations.

The companion submission to your manuscript has also been evaluated on its merits, and we have reached a different decision based on the reviewers' recommendation. Therefore, we cannot guarantee to publish the two submissions side-by-side unless you choose to delay publication of this submission. 

Sincerely,

Chaoyang Xue, Ph.D.

Academic Editor

PLOS Pathogens

Alex Andrianopoulos

Section Editor

PLOS Pathogens

Michael Malim

Editor-in-Chief

PLOS Pathogens

orcid.org/0000-0002-7699-2064

Reviewer Comments (if any, and for reference):

Reviewer's Responses to Questions

**Part I - Summary**

Reviewer #1: In this paper, Xiong and colleagues investigate the role of ERG251, due to its conserved regulatory relationship with the Efg1 transcription factor in multiple strains of C. albicans. As Efg1 controls biofilm formation in multiple strains, the authors demonstrate that Erg251 is a critical part of that regulation. The phenotype of the erg251 strain is only revealed under biofilm conditions, with accumulation of alternate sterols in the membrane under hypoxia. The authors also demonstrate that Erg25, a duplicated gene, can partially compensate for loss of Erg251 in biofilm formation, but only when it is overexpressed.

This is a clear and well written paper. My main concern is about the model in figure 7, which I think needs additional data.

Reviewer #2: This manuscript presents a very thorough analysis of the role of the Erg251 gene in controlling biofilm-specific ergosterol regulation during biofilm formation and shows that this gene is necessary for biofilm formation, but not for planktonic growth. The authors study this gene's role in multiple C. albicans strains and find similar results throughout. They demonstrate that loss of Erg251 leads to biofilm-specific loss of ergosterol and accumulation of intermediates. They also demonstrate that the Erg25 parlog does not play a role in this phenotype, but if over expressed it can partially complement loss of Erg251. The one minor weakness that I noted was they measured Erg25 overexpression in Figure S2c, but only in planktonic conditions. Biofilm conditions would be more relevant given the emphasis of this relationship in the manuscript,

Reviewer #3: This paper approaches the question of why Candida albicans and related yeasts have two genes for C-4 sterol methyl oxidases (Erg25 and Erg251) while other fungi and algae can get by with only one. The authors do a thorough and convincing job of showing that ERG251 is required for growth under partially anaerobic conditions (biofilms) but not under fully aerobic conditions (planktonic cells). This observation expands on Candida's ability to thrive under many different growth conditions requiring metabolic flexibility. These growth studies are also integrated with changes in fluconazole sensitivity, which are interpreted appropriately in terms of changing levels of carbon flow through slightly altered paths for sterol biosynthesis. That's all to the positive. The following are items that might be addressed for improved clarity and impact.

1/ The paper mentions planktonic versus biofilm conditions repeatedly, and the Discussions focused on this distinction almost exclusively. Yet your standard conditions are really only comparing cells grown in YPD with shaking versus YPD sealed without shaking, with the latter being anaerobic to varying degrees as growth proceeds. I have no problem with the practicality of this distinction, and you cite references for the strong overlap of genes turned on for hypoxia and biofilms. It would be nice if you made an effort to distinguish hypoxia/anaerobic genes from biofilm genes. Would your mutants defective in erg25 or erg251 grow in liquid anaerobic conditions, such as those described by Dumitru et al AAC 2004?

2/ The theme that ERG251 provides C. albicans with greater flexibility under anaerobic conditions (gut) could easily be tested if you switched from YPD to a chemically defined growth medium (RPMI??). Andreasen and Stier (J Cell Physiol 1953) showed that S. cerevisiae had a nutritional requirement for ergosterol anaerobically. Please add this reference to line 303. When looking at similar conditions for C. albicans, Dumitru et al expected that Candida too would require ergosterol, but it did not. They were unable to provide a good explanation for how C. albicans differed in this regard. Now you could suggest that having both ERG251 and ERG25 provides the answer, and you have the mutants to prove it.

3/ I like your experiments Figure 5a and S7 exploring the sterols produced under your "biofilm" conditions, but it troubled me to see them left as Sterol A and sterol B and later referred to as "novel" instead of not yet identified. They do not match eburicol or 4,4 dimethyl zymosterol. But Erg 25 and Erg251 are 4-methyl oxidases, and in other organisms there are other potential substrates that could accumulate in the erg251 double mutant. Ergosterol is also the dominant sterol in Chlamydomonas reinhardtii (Brumfield, Eur J Phycol 2017) and Chloella variabilis (Voshall, J Phycol 2021). These pathways to ergosterol differ from the yeasts in some significant ways but significantly, in one putative sterol pathway, four sterols are alternate substrates for a single C-4 methyl oxidase, three of which seem possible for C. albicans grown under hypoxic conditions: obtusifoliol, 24-dihydroobtusifoliol, and 24 methylene lophenol. Please forgive me if you have already made extensive efforts to identify these sterols and they are in fact novel. I would love to see Fig S7 incorporated into Fig 5a.

**Part II – Major Issues: Key Experiments Required for Acceptance**

Reviewer #1: The model in figure 7 suggests that both Erg25 and Erg251 are well expressed under planktonic conditions and that there is compensation that cannot be achieved in biofilm conditions when Erg251 is lost. In Figure S2, it looks like Erg251 is the major paralog in both conditions, as its expression is higher than Erg25 in both. If anything, the relative abundance is much higher in planktonic, so the phenotype of the mutant should be more apparent in planktonic conditions. Figure legend 7 says “higher expression levels of ERG251 relative to ERG25 may contribute to differences in function of the two paralogs under biofilm conditions” is not well supported by the current data. The argument that biofilm conditions limit ergosterol biosynthesis should be further supported by comparing ergosterol levels in the WT between conditions, and by analyzing ERG25 and ERG251 transcripts in the WT between conditions. Additionally, the authors should make and characterize the Erg25 Erg251 double mutant strain.

The authors should also perform a standard CLSI methods test for fluconazole susceptibility of the different mutant strains. As C. albicans grows well in 96-well plates, this would allow for a clearer and more quantitative measure of fluconazole susceptibility.

Reviewer #2: There are no major issues.

Reviewer #3: None

**Part III – Minor Issues: Editorial and Data Presentation Modifications**

Reviewer #1: The alternate sterols produced by the mutants are quite interesting, and I would recommend moving SFig7 a into the main figure 5.

Reviewer #2: I think they should measure ERG25 expression in the epistasis strains in biofilm conditions.

Reviewer #3: My feeling is that the paper is longer than needed for the points being made, but since I have no concrete suggestions on what should be cut, I haven't said so in my comments to the authors. I mention it only in case the other reviewer(s) do make such suggestions.

PLOS authors have the option to publish the peer review history of their article (what does this mean?). If published, this will include your full peer review and any attached files.

Reviewer #1: No

Reviewer #2: No

Reviewer #3: **Yes: **Kenneth W Nickerson

Figure Files:

Data Requirements:

Reproducibility:

References:

---

## [Decision Letter · Decision Letter 1]

25 Apr 2024

Dear Professor Mitchell,

We are pleased to inform you that your manuscript 'Biofilm-associated metabolism via ERG251 in Candida albicans' has been provisionally accepted for publication in PLOS Pathogens.

In addition, you may include the information that the erg25 erg251 mutant is essential in the manuscript during formatting change, as one reviewer recommended.

Best regards,

Chaoyang Xue, Ph.D.

Academic Editor

PLOS Pathogens

Alex Andrianopoulos

Section Editor

PLOS Pathogens

Michael Malim

Editor-in-Chief

PLOS Pathogens

orcid.org/0000-0002-7699-2064

Reviewer Comments (if any, and for reference):

Reviewer's Responses to Questions

**Part I - Summary**

Reviewer #1: The authors have fully addressed all of my concerns. The only thing that I would add is that the erg25 erg251 mutant being essential would be of interest to the community, so I would recommend that the authors add this information to the text. I can imagine a situation in which others in the field, who are potentially less familiar with genetic transformation, would try and generate this strain and fail, and not be confident in their results.

Reviewer #2: (No Response)

Reviewer #3: This paper has been suitably modified.

**Part II – Major Issues: Key Experiments Required for Acceptance**

Reviewer #1: (No Response)

Reviewer #2: (No Response)

Reviewer #3: None

**Part III – Minor Issues: Editorial and Data Presentation Modifications**

Reviewer #1: (No Response)

Reviewer #2: (No Response)

Reviewer #3: None

PLOS authors have the option to publish the peer review history of their article (what does this mean?). If published, this will include your full peer review and any attached files.

Reviewer #1: No

Reviewer #2: No

Reviewer #3: **Yes: **Kenneth W Nickerson

---

## [Editor Report · Acceptance letter]

6 May 2024

Dear Professor Mitchell,

We are delighted to inform you that your manuscript, "Biofilm-associated metabolism via *ERG251* in *Candida albicans*," has been formally accepted for publication in PLOS Pathogens.

Best regards,

Michael Malim

Editor-in-Chief

PLOS Pathogens

orcid.org/0000-0002-7699-2064